# Forecasting daily new infections, deaths and recovery cases due to COVID-19 in Pakistan by using Bayesian Dynamic Linear Models

**Firdos Khan** [1]*, **Shaukat Ali**[2], **Alia Saeed**[3,4], **Ramesh Kumar**[3], **Abdul Wali Khan**[5]

**1** School of Natural Sciences (SNS), National University of Sciences and Technology (NUST), Islamabad, Pakistan, **2** Global Change Impact Studies Centre (GCISC), Ministry of Climate Change, Islamabad, Pakistan, **3** Health Services Academy, Islamabad, Pakistan, **4** ClimatExperts, Islamabad, Pakistan, **5** Ministry of National Health Services, Regulations and Coordination Islamabad, Islamabad, Pakistan

* fkyousafzai@gmail.com

**Data Availability Statement:** All relevant data are within the manuscript and its Supporting Information files.

## Abstract

The COVID-19 has caused the deadliest pandemic around the globe, emerged from the city of Wuhan, China by the end of 2019 and affected all continents of the world, with severe health implications and as well as financial-damage. Pakistan is also amongst the top badly effected countries in terms of casualties and financial loss due to COVID-19. By 20th March, 2021, Pakistan reported 623,135 total confirmed cases and 13,799 deaths. A state space model called 'Bayesian Dynamic Linear Model' (BDLM) was used for the forecast of daily new infections, deaths and recover cases regarding COVID-19. For the estimation of states of the models and forecasting new observations, the recursive Kalman filter was used. Twenty days ahead forecast show that the maximum number of new infections are 4,031 per day with 95% prediction interval (3,319–4,743). Death forecast shows that the maximum number of the deaths with 95% prediction interval are 81 and (67–93), respectively. Maximum daily recoveries are 3,464 with 95% prediction interval (2,887–5,423) in the next 20 days. The average number of new infections, deaths and recover cases are 3,282, 52 and 1,840, respectively, in the upcoming 20 days. As the data generation processes based on the latest data has been identified, therefore it can be updated with the availability of new data to provide latest forecast.

## 1. Introduction

Various pandemics and contagious viral infections such as influenza, Zika, MERS, Spanish flu, Ebola emerged in the past, which badly affected the human lives and economy of the of the major areas and regions of the world [1, 2]. Currently the world is facing a viral infectious disease caused by severe acute respiratory syndrome corona virus 2 (SARS-CoV-2), initially reported in the Wuhan city of China in December 2019, spreading across all continents of the world and it was named as COVID-19 [3]. With a time span of three months, this virus spread rapidly through enormous ways and reached to most countries of the world [3]. The world came up with different measures to contain the spread of the virus such as limiting the social mobility and lockdown from time to time and consequently reached to the peak of inclined economical losses. The second wave of COVID-19 was more lethal due to more deaths, in

**Funding:** The authors received no specific funding for this work.

**Competing interests:** The authors have declared that no competing interests exist.

comparison to the first wave. The third wave has begun in Pakistan which is more alarming than the second wave of COVID-19 in terms of new emerging infections, contagiousness, complications, hospital admission and daily reported-fatalities. According to World Health Organization (WHO) and Worldometer statistics about COVID-19, Pakistan is at 31st position in the world based on the total confirmed cases [3, 4].

The time series models (TSMs) have multitude of applications and play significant role in the areas like finance, economics, engineering, climatology, epidemiology, and hydrology in terms of forecasting [5–10]. TSMs do not merely describe the existing trends but can help to explain the data generating process (DGP) and predict future trends. Moreover, TSMs are advantageous over mechanistic models (MMs) to forecast future disease trends due to the highly explicit epidemiological information, which is needed to fit MMs [10].

Most of TSMs including the popular classes, namely, autoregressive (AR), moving average (MA), autoregressive moving average (ARMA), autoregressive integrated moving average (ARIMA) are useful to deal with the stationary data. Therefore, if the data is not stationary, it is necessary to transform the data so that it becomes stationary. Such transformations could have impacts on the results. In many cases, results in original units are required and thereafter the results are needed to be retransformed. State space models (SSMs) offer a very rich class of models which are advantageous over the above mentioned TSMs such as; no need of stationarity and thus no need of transformation of the data [11–16]. SSMs consider a time series data as the output of a system which is dynamic and perturbed by random disturbances. Bayesian dynamic linear model (BDLM) or simply dynamic linear model (DLM) is a special case of general SSMs, i.e., it is linear and Gaussian and useful to model and forecast time series data. In DLMs, estimation of states and forecasting can be done recursively by utilizing the well-known Kalman filter [11]. These models allow a natural explanation of a time series data as the combination of several components, such as seasonal, trend or regressive components. In addition, they have the powerful and sophisticated probabilistic structure, and thus presenting an adaptable framework for a very wide variety of applications in different areas. Given the information available, computations including estimation and forecasting can be done utilizing recursive algorithms by computing the conditional distribution of the quantities of interest. Therefore, naturally these models are treated within a Bayesian framework. For further details about DLMs and their applications, refer is made [11–16].

In the current pandemics, to model and forecast new infections, deaths and recoveries due to COVID-19, different researchers have used different techniques including statistical, mathematical, machine learning algorithm, deep learning etc. [17–21]. [21] compared six time series models, including ARIMA [7], the Holt–Winters additive model (HWAAS) [22], TBAT [23], Facebook's Prophet [24], DeepAR [25] and N-Beats [26]. They concluded that ARIMA and TBAT models performed better for seven countries out of ten countries. [27] compared different forecasting methods to choose the best method for forecasting deaths due to COVID-19 in the world. These methods include simple average, moving average, naive method, Holt linear trend method, single exponential smoothing, ARIMA and Holt-Winters method. They concluded that ARIMA provided the best-fit model to their time series data. [28] used Holt model and ARIMA model to predict the future's situation of COVID-19 in China and the United State of America (USA). They concluded that the epidemic situation has ended after May in the Hubei Province of China, however, the situation in the USA has become more sever after May 2020. [29] used a hybrid approach by combining fractals and fuzzy logic to forecast COVID-19 in ten countries with the forecast windows of 10 and 30 days. [30] made a comparison between multiple ensemble neural network model with fuzzy response aggregation and monolithic neural networks and concluded that the former outperformed the later one. [31] used multivariate Markov prognostic models to identify high-mortality risk in hospitalized

COVID-19 patients, however, these models require different data sets including comorbidities, demographics, and laboratory values taken at admission and during hospitalization.

[18] used ARIMA model to forecast new infections, death and recover cases due to COVID-19 in Pakistan. Their results revealed high exponential growth in the number of new infections, deaths and recover cases during May 2020 in Pakistan. [32] used vector Autoregressive (VAR) models to model and forecast daily new cases, deaths, and recoveries with respect to COVID-19 in Pakistan. The results of [18] have more deviations from the real data, however, the findings of [32] are closer to the recorded data. [19] compared different forecasting models for cumulative new cases and recover cases for the duration of February and June 2020. Their results concluded that ARIMA model performed better than the other models. Most of the time series models (ARIMA, SARIMA, AFRIMA, and non-linear models including ARCH, GARCH etc.) required stationary time series data, however, by applying a transformation on data, we may lose important information regarding extreme phenomena. DLM does not requires the assumption of stationarity and it can be used to model and forecast a time series data with no stationarity, structural breaks and no-clear pattern. It can be seen that none of the studies used DLM to forecast COVID-19 in Pakistan. Secondly, most of the mentioned studies in the context of Pakistan had 5 days or 10 days ahead forecast for new infections, deaths and recover cases. Due to the flexible framework and dynamic nature, we propose to use DLM for modelling and forecasting daily new cases, deaths and recoveries regarding COVID-19 in Pakistan. In addition, this study will provide forecast for longer duration than the previous studies which may help concerned department in making their policies.

This study aims to identify the data generating process of three variables (daily new infections, deaths and recover cases), using DLM; to provide forecast about above mentioned three variables due to COVID-19 in Pakistan.

## 2. Data and study area

Diurnal data of COVID-19 was collected for day-to-day new cases, deaths, and recover cases from World Health Organization (WHO) and Worldometer [3, 4]. The 1st case of COVID-19 was reported on 26th February 2020 in Pakistan. Therefore, the data used in this study ranges between February 26, 2020 to March 20, 2021. Our study region is Pakistan, nevertheless, the proposed methods can be used to the subregions or larger areas.

## 3. Methodology

### 3.1. DLM's structure

DLMs is a popular technique for smoothing and forecasting time series. It has two main components, unobserved states and observed data. The model is explained for daily new infections (DNI) only, however, it can be explained in the same way for daily deaths and recover cases.

unobserved states: $\theta_1, \theta_1, \ldots, \theta_t$

observed observation: $DNI_1, DNI_2, \ldots, DNI_t$

where $DNI_t$ represents daily new infections of COVID-19 in this study. The $P(\theta_t/\theta_{t-1})$ is the transition probability of states implying the well-known Markov property where the probability of current state depends only on the previous state for $t = 1,2,\ldots,T$. The probability of observed data $DNI$ at time $t$ is $P(DNI_t/\theta_t)$ implying that observed data depend on the current state. The models can be formulated as:

$$\theta_t/\theta_{t-1} \sim N(G_t \theta_{t-1}, W_t)$$

$$DNI_t/\theta_t \sim N(F_t \theta_t, E_t)$$

Then the DLM can be represented in the following equations:

$$DNI_t = F_t\theta_t + \epsilon_t \qquad \varepsilon_t \sim N(0, E_t) \tag{1}$$

$$\theta_t = G_t\theta_{t-1} + \omega_t \qquad \omega_t \sim N(0, W_t) \tag{2}$$

The $\epsilon_t$ and $\omega_t$ are two independent white noise error terms which are independent both within each other and between them with zero mean and known covariances $E_t$ and $W_t$, respectively. Eqs (1) and (2) are called observation equation and state or system equation, respectively. It is further assumed that the prior distribution of $\theta_0$ is Gaussian distribution, i.e.,

$$\theta_0 \sim N(m_0, C_0)$$

where $\boldsymbol{\theta_t}$ is a vector of unobserved states of the system of length m that are assumed to evolve over time according to the linear system operator $\boldsymbol{G_t}$ (state transition), a matrix of order $m{\times}m$. For time series data the states or different features can be trend, seasonality or regressive components [12, 33]. Then the observations can be expressed as in Eq (1). We observe a linear combination of the states with a matrix $\boldsymbol{F_t}$ ($m{\times}p$) which serves as observation operator that transforms the model states to a time series observation. The dependence structure of the model presented in Eqs (1) and (2) is given below:

$$\theta_0 \rightarrow \theta_1 \rightarrow \theta_2 \rightarrow \ldots \rightarrow \theta_{t-1} \rightarrow \theta_t \rightarrow \theta_{t+1} \rightarrow \ldots$$
$$\downarrow \quad \downarrow \qquad\quad \downarrow \qquad \downarrow \qquad \downarrow$$
$$DNI_1 \quad DNI_2 \quad \ldots \quad DNI_{t-1} \quad DNI_t \quad DNI_{t+1}$$

## 3.2. State's estimation of DLM

For a given DLM, the major tasks are to draw inference about the unobserved states or to forecast future observations based on a part of available observation sequence [34]. Conditional distributions of the quantities of interest given the available information are used to solve the problem of estimation and forecasting. To estimate the state's vector, we compute the conditional probability density $p(\theta_s|DNI_{1:t})$. It is imperative to differentiate among filtering ($s = t$), state prediction ($s < t$) and smoothing ($s > t$). In filtering the data is assumed to arrive sequentially which is usual in time series. Now there is need a procedure to estimate the current value of the state, on the basis of observation up to time t (for example now), and to update our estimate and forecast as the new data become available for the next time ($t+1$). To solve the problem of filtering, we compute the conditional density $p(\theta_t|DNI_{1:t})$. In DLM, the Kalman filter [35] provide formulae to update our current inference on the state vector as the new information become available. This refers to passing form the filtering density $p(\theta_t|DNI_{1:t})$ to $p(\theta_{t+1}|DNI_{1:t+1})$.

## 3.3. Validation of DLM

The fitted DLM can be validated graphically as well as numerically. Graphically the performance of the fitted model can be assessed by analyzing the residuals and checking their histogram, probability density function and quantile-quantile plot (qqplot) of observed and model's simulated data. The residuals of the fitted model should have Gaussian distribution with zero mean value if the model is properly specified. Also, a comparison can be made between the observed and model predicted data to assess the performance of fitted model. The performance of the fitted model is evaluated numerically by comparing the distribution of observed and simulated data which includes minimum values, first quartile, mean, median, third quartile and maximum values. In addition, the model can be evaluated by comparing the probability density function of observed data with model's simulated data.

### 3.4. Forecasting with DLM

Forecasting is the eventual objective of time series modelling and the length of forecasting depends on the nature and objectives of the study. The estimation of state is then just a step for the prediction of future's observation. For instance, if we wish one step ahead forecasting of next observation $DNI_{t+1}$ based on available data $DNI_{1:t}$. Then first we need to estimate the next value of state vector $\theta_{t+1}$ and then forecast the next observation based on the state $\theta_{t+1}$. The one step ahead predictive density of the state is $p(\theta_{t+1}|DNI_{1:t})$ which based on the filtering density of $\theta_t$. Then consequently, from this the predictive density of observation can be calculated as $p(DNI_{t+1}|DNI_{1:t})$. If the interest is k-step ahead forecasting about DNI ($DNI_{t+k}$), then we need to estimate the evaluation of the system denoted by $\theta_{t+k}$. The predictive density $p(\theta_{t+k}|DNI_{1:t})$ can be used to solve the state prediction. Once the predictive density of k-step ahead state is obtained then based on this density, the k-step ahead predictive density $p(DNI_{t+k}|DNI_{1:t})$ of the future observation can be calculated at time $t+k$. The forecast become more and more uncertain as the forecast duration $t+k$ get further away in future, however, the uncertainty can be quantified by using the predictive density of $DNI_{t+k}$ given $DNI_{1:t}$ [12, 36]. The above procedure can be repeated in the same fashion for daily deaths and recover cases due to COVID-19 in Pakistan by replacing DNI by DD (daily deaths) and DRC (daily recover cases), respectively.

## 4. Results

The results of the study are divided and presented into three subsections:

### 4.1. Model's evaluation

The specified DLMs are evaluated graphically and as well as numerically in Figs 1–3 and Table 1. Figs 1–3 show the probability density function of daily new cases, deaths and recover cases both for observed and models' simulated data. Lines are drawn at 25th, 50th, 75th and 95th percentiles in each graph both for observed and simulated data to make it easy in understanding. The results show that the fitted models closely simulated the observed data sets, however, the difference between observed and simulated data is little higher at 95th percentiles. The 95th percentile is overestimated for all three variables, but the difference is smaller for daily new

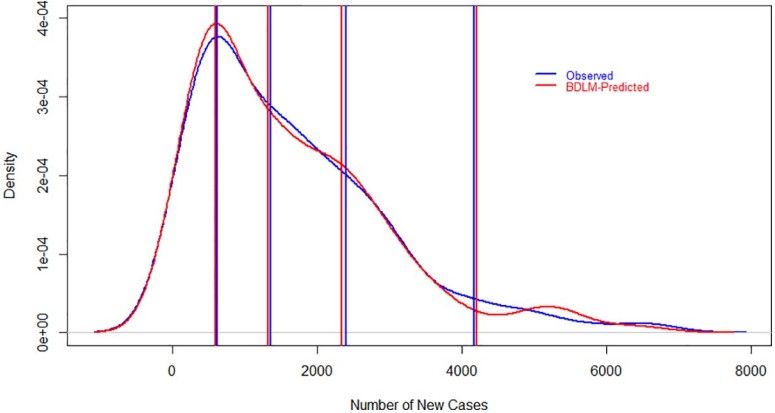

**Fig 1. Evaluation of DLM using probability density function of observed and model simulated daily new cases of COVID-19 in Pakistan.** The vertical lines show 25th, 50th, 75th and 95th percentiles for observed and simulated data sets. Red and blue colors represent observed and simulated data sets, respectively. On the x-axis and y-axis, the number of daily new cases and density are given, respectively.

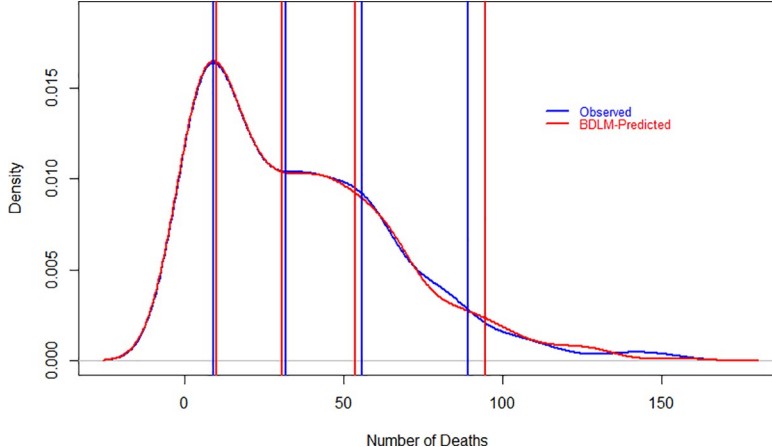

**Fig 2. Evaluation of DLM using probability density function of observed and model simulated daily deaths of COVID-19 in Pakistan.** The vertical lines show 25th, 50th, 75th and 95th percentiles for observed and simulated data sets. Red and blue colors represent observed and simulated data sets, respectively. On the x-axis and y-axis, the number of daily deaths and density are given, respectively.

cases as compared to daily recover cases. The model well captured the daily deaths rather than new and recovered cases, which could be a good estimate of mortality rates due to COVID-19. Table 1 has numerical evaluation about fitted DLMs for all considered three variables by comparing the distribution of observed and simulated data. For daily new cases, the models captured well the distribution of observed data where minimum values are the same for observed and simulated data sets. However, the remaining statistics in Table 1 are slightly underestimated for daily observed cases. For daily deaths, the statistics related to simulated and observed data are very close. Observed and simulated median, 3rd quartile and maximum values are 32, 56, 153 and 31, 54, 155, respectively. Regarding daily recover cases, the differences

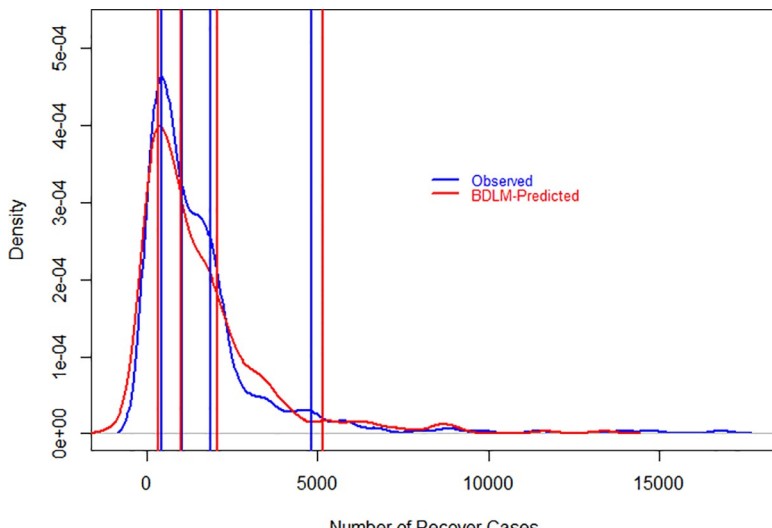

**Fig 3. Evaluation of DLM using probability density function of observed and model simulated daily recover cases of COVID-19 in Pakistan.** The vertical lines show 25th, 50th, 75th and 95th percentiles for observed and simulated data sets. Red and blue colors represent observed and simulated data sets, respectively. On the x-axis and y-axis, the number of daily recover cases and density are given, respectively.

**Table 1. Distribution of observed and simulated daily new cases, daily deaths, and daily recover cases of COVID-19 in Pakistan.**

| Variable | | Min. | 1st Quar. | Median | Mean | 3rd Quar. | Max |
|---|---|---|---|---|---|---|---|
| Daily new infections | Observed | 0 | 617 | 1352 | 1649 | 2397 | 6825 |
| | Simulated | 0 | 589 | 1318 | 1622 | 2337 | 6681 |
| Daily Deaths | Observed | 0 | 9 | 32 | 36 | 56 | 153 |
| | Simulated | 0 | 10 | 31 | 35 | 54 | 155 |
| Daily Recover Cases | Observed | 0 | 411 | 1013 | 1534 | 1855 | 16813 |
| | Simulated | 0 | 331 | 978 | 1527 | 2042 | 13375 |

are little higher as compared to daily new cases and daily deaths. However, the model closely reproduced the minimum value, median and mean values which are 0, 1,013, 1,534 and 0, 978, 1,527, respectively, for observed and simulated data.

## 4.2. Diagnostic checking

The results about diagnostic checking of the models are presented in Figs 4–6. Figs 4–6 are about daily new cases, deaths and recover cases, respectively. The first panel (top-left) of Fig 4 shows residuals plot and it can be seen that it is centered on zero. The histogram (top-right) and pdf (bottom-left) show that the residuals of the specified model are approximately normally distributed. The fourth panel (bottom-right) of Fig 4 presents the quantile-quantile plot of observed and model's simulated daily new cases. For daily deaths and daily recover cases, the diagnostic checking results are given in Figs 5 and 6, respectively. Once the fitted models qualify the diagnostic checks, then it can be used for forecasting.

## 4.3. Forecasting

The forecasting results about daily new cases, deaths and recover cases are presented in Figs 7–9, respectively. Fig 7 shows a comparison between observed and model simulated data for the duration of March 2020 to March 2021 where it can be noted that how closely the specified model reproduced the observed data. On the right side of the vertical line, the results are about forecasting of daily new cases with their 95% prediction interval. In the forecasting duration, DLM captured the variability of the data very well. This is one of the reasons that is why DLMs

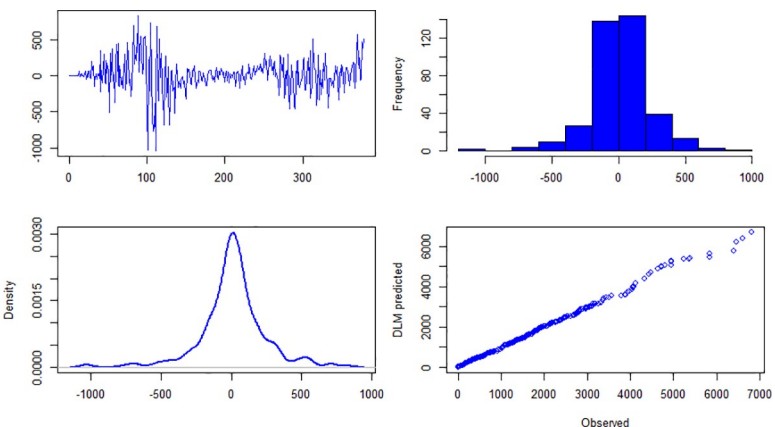

**Fig 4. Upper left panel is residuals, upper right panel is histogram of residuals, bottom left panel is probability density function of residuals, bottom right panel is qqplot of observed and model simulated daily new cases of COVID-19 in Pakistan.**

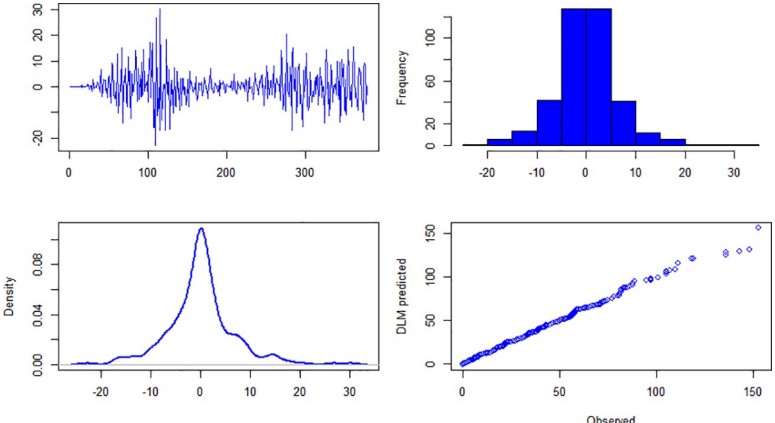

**Fig 5. Upper left panel is residuals, upper right panel is histogram of residuals, bottom left panel is probability density function of residuals, bottom right panel is qqplot of observed and model simulated daily deaths of COVID-19 in Pakistan.**

are prefer over other time series model because DLMs can elegantly model a time series with non-stationarity nature, structural breaks, no clear pattern etc. Nevertheless, the model captured well the variability, but a 95% prediction interval was calculated to encounter the uncertainty. The horizontal line shows the average value of observed daily infections due to COVID-19. It is clear from Fig 7 that the forecasting daily new cases are higher than the average value of new cases in Pakistan. Table 2 summarized the forecasting results. It can be seen that the minimum daily new cases in the forthcoming 20 days are 2,479 with prediction interval of 1,767–3,191. The maximum number of daily new cases are 4,031 with 95% prediction interval of 3,319–4,743. These results indicate that that the average number of daily new cases in the upcoming 20 days is 3,282 with 95% prediction interval of 2,570–3,994. The model's forecast depicts that there will be 65,638 total new infections in the next 20 days.

Fig 8 shows the results of daily deaths due to COVID-19 in Pakistan. The results on the left side of the vertical line in Fig 8 shows a comparison between observed and model's simulated

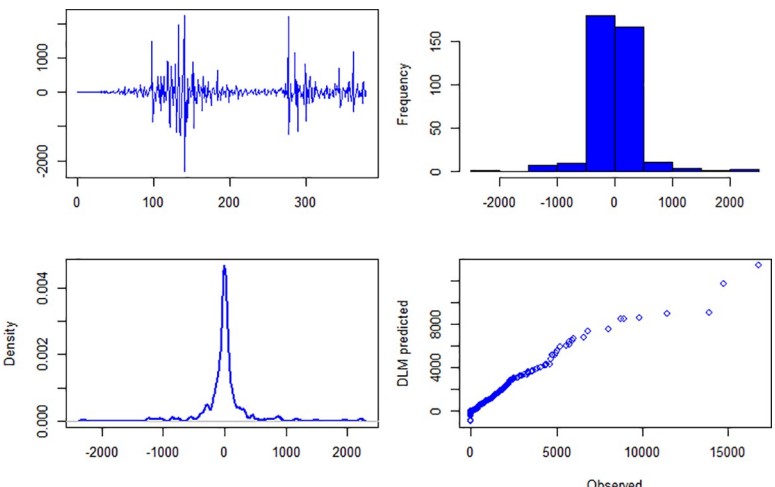

**Fig 6. Upper left panel is residuals, upper right panel is histogram of residuals, bottom left panel is probability density function of residuals, bottom right panel is qqplot of observed and model simulated daily recover cases of COVID-19 in Pakistan.**

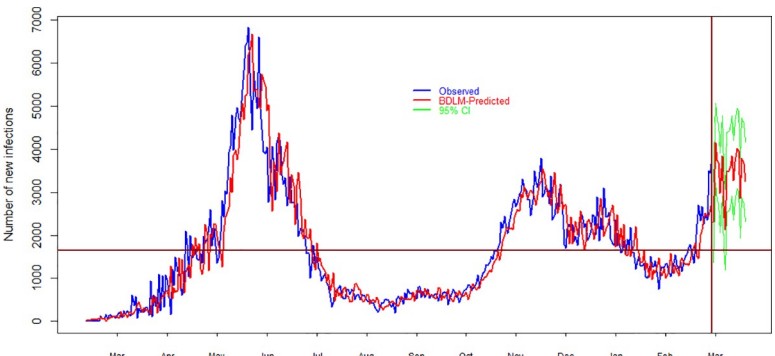

**Fig 7. Comparison of observed and model simulated (average of 100 simulations) daily new cases of COVID-19 in Pakistan.** Blue, red and green colors show observed, model simulated and 95% prediction intervals of daily new cases. Horizontal line shows the average value of observed data while the vertical line shows the demarcation points between forecast and observed data. On the x-axis and y-axis, time in months and the number of daily new cases are given, respectively.

daily deaths during March 2020 to March 2021. On the right side of the vertical line, the results are about forecasting daily deaths and their 95% prediction interval. It is clear that the model elegantly captured the structure of the time series. It can be noted and that the number of deaths is higher than the average value during the forecast duration where the horizontal line indicates the average value of observed deaths. A summary of the model's forecast is given in Table 2. Table 2 shows that the minimum number of deaths are 42 with 95% prediction interval of 29–56. The maximum number of forecast deaths during the upcoming twenty days is 81 with 95% prediction interval of 67–93. The forecast results indicate that on the average, there are 52 deaths during next twenty days with 95% prediction intervals of 38–65. In addition, the forecasting results show that the total number of deaths during the upcoming 20 days is 1,035.

Fig 9 presents the results for daily recover cases about COVID-19 in Pakistan. Fig 9 shows a comparison between observed and model simulated daily recover cases during March 2020 and March 2021. On the right side of the vertical line, the results are about model's forecasting where the red line is the forecast daily new cases and green lines show their 95% prediction interval. The forecasting results suggest that the recover cases in the upcoming 20 days are little higher than the average value of observed daily recover cases. The forecast results about daily

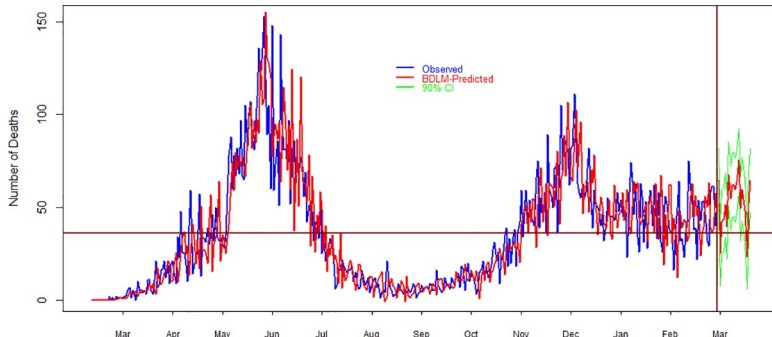

**Fig 8. Comparison of observed and model simulated (average of 100 simulations) daily deaths of COVID-19 in Pakistan.** Blue, red and green colors show observed, model simulated and 95% prediction intervals of daily deaths. Horizontal line shows the average value of observed data while the vertical line shows the demarcation points between forecast and observed data. On the x-axis and y-axis, time in months and the number of daily deaths is given, respectively.

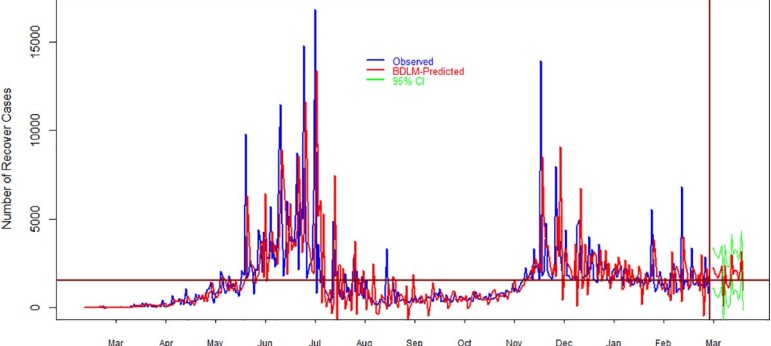

**Fig 9. Comparison of observed and model simulated (average of 100 simulations) daily recover cases of COVID-19 in Pakistan.** Blue, red and green colors show observed, model simulated and 95% prediction intervals of daily recover cases. Horizontal line shows the average value of observed data while the vertical line shows the demarcation points between forecast and observed data. On the x-axis and y-axis, time in months and the number of daily recover cases are given, respectively.

recover cases are summarized in Table 2. The results suggest that the minimum number of daily recover cases during the next twenty days is 578 with 95% prediction intervals of 0–2,536. The maximum number of daily recover cases is 3,464 with 95% prediction intervals of 2,887–5,423. However, the forecast's results suggest that the average number of daily recover cases in the forthcoming twenty days is 1,840 with 95% prediction intervals of 1262–3,799. The results further indicates that the total number of daily recover cases during the next 20 days are 36,784.

# 5. Discussion

COVID-19 has changed the lifestyle, businesses, education system and many more around the world since it emerged in the end of 2019 [37] and it is being considered that the COVID-19 pandemic is the most significance global crises after world war II [38]. However, the long-term impacts depend on the persistence of the pandemic. Due to alarming spread, surge in cases, complications and daily reported morbidity and mortality indicators, this pandemic is the focus of researchers and tried to forecast daily cases (infections, deaths, and recover cases) due to COVID-19. For instance [18], used ARIMA model to forecast the future spread of daily new cases, deaths and recover cases. Their results suggest that the new infections will increase by

**Table 2. Forecast values for daily infections, fatalities and recover cases about COVID-19 with their corresponding 95% confidence intervals for Pakistan in upcoming 20 days.** The forecast results based on the average of 100 simulations.

| Variable | Max/Min | Forecast | 95% Confidence Intervals | | Total Cases in 20 days |
| --- | --- | --- | --- | --- | --- |
| | | | Lower limit | Upper Limit | |
| Infections | Min | 2479 | 1767 | 3191 | 65,638 |
| | Max | 4031 | 3319 | 4743 | |
| | Average | 3282 | 2570 | 3994 | |
| Fatalities | Min | 42 | 29 | 56 | 1,035 |
| | Max | 81 | 67 | 93 | |
| | Average | 52 | 38 | 65 | |
| Recover cases | Min | 578 | 0 | 2536 | 36,784 |
| | Max | 3464 | 2887 | 5423 | |
| | Average | 1840 | 1262 | 3799 | |

2.7 times and number of deaths to eightfold by May 2020. However, besides forecast's error the government policies affected the spread of COVID-19 and did not touch these high values. [39] used ARIMA with Kalman Filter to model and forecast the future behavior of COVID-19 in Pakistan. [39] made forecast for the first five days of May 2020 based on the available data. Their results suggest that the new infections, deaths and recoveries will be reached to 15,652, 6,342 and 516, respectively. The results of [39] have an increasing trend in all variables and provide higher estimates for maximum values than [18]. [19] used ARIMA model for modelling and forecasting cumulative number of confirmed cases, deaths and recover cases in Pakistan. Their study provides forecast for only ten days (25 June to 4 July 2020). Findings of their study suggest that the cumulative number of new infections, deaths and recover cases would be 2,31,239, 5043 and 1,11,616, respectively, at the end of forecast horizon. [32] used VAR model for forecasting future scenarios about COVID-19 and provides 10 days ahead forecast (28 June to 7 July 2020). The results of their study suggested that the maximum number of daily new infections, deaths and recover cases would be 5,363, 167 and 4,016 during the forecast duration. [29] combined fuzzy logic and fractal dimension to model and forecast time series of COVID-19 (confirmed cases and deaths). They used forecasting windows of 10 and 30 days with the forecast accuracy of 98%. [30] used ensemble neural network model with fuzzy response aggregation to predict COVID-19 time series in Maxico. [40] used differential equation model to model and forecast future's situation under different assumption. Based on their results, they recommended to take strong controlled measured for infected and asymptotic patients to reduce the number of total infections in the future. The current study provides twenty days ahead forecast (21 March to 9 April 2021) based on available data. The forecast results suggest that the minimum and maximum number of new infections are 2,479 and 4,031, respectively. The minimum and maximum daily deaths are 42 and 81, respectively, during the forecast period. Minimum and maximum daily recover cases are 578 and 3,464, respectively during the next twenty days. The results suggests that the total number of new infections, deaths and recover cases during the entire forecast duration are 65,638, 1,035 and 36,784, respectively.

It can be observed that the results of this study are stable and are consistent with the trends of considered variables. The possible reasons may include: the techniques we used consider the dynamic nature of the system; secondly, more observations have been used in this study and perhaps this also exaggerated the findings. It is imperative to state that forecast is a complicated subject, therefore, these results could change due to various reasons including government policies regarding the current or future situation of COVID-19. This may include lockdown, closure of institutions or reducing the number of employees in a day at offices etc. The results of [41] suggested that 15 days after the lockdown, daily new infections due to COVID-19 and growth factor of this disease showed decreasing trend, however, there was no significance decline in the mortality and prevalence in 27 randomly selected countries. [42] investigated the impacts of lockdown and social distancing with deaths due to COVID-19 in 16 European countries. Their results suggested that there was close relationship between the deaths due to COVID-19 and the days elapsed until lockdown. However, there was week relationship between deaths and social distancing. There are other studies which investigated that screening, quarantine, isolation in different settings and contact tracing can help in reducing the new infections due to COVID-19 [43, 44].

## 6. Conclusion and recommendations

The forecast findings of the study indicate that the average daily new cases are higher than the average values of the observed data. The maximum and minimum number of daily new cases

during the next twenty days are 2,479 and 4,031, respectively. The average number of daily new cases and the total number of daily new cases during the forecasting period are 3,282, and 65,638, respectively. The results of the daily deaths show that the minimum and maximum numbers are 42 and 81 per day, respectively. Average daily deaths during the upcoming twenty days are 52. The forecast results advocate that the total number of deaths during the next twenty days are 1,035. The forecast results for daily recovery cases demonstrate that on the average there are 1,840 recoveries per day during the next twenty days. The minimum and maximum number of daily recover cases during the forecasting period are 578 and 3,464, respectively. The total number of daily recovery cases during the upcoming twenty days are 36,784.

The findings of this study may be helpful to epidemiologist, to design future modeling based on this evidence and also for policy-makers, planners, and managers in health sector. The specified models can be updated with the arrival of new data and therefore, this forecast could be used on regular basis to provide rigorous information for decision making by the relevant departments in Pakistan. Recommendations about future's research include Bayesian time series modelling and forecasting of daily new infections, deaths and recover cases, however, prior information and its quantification may not be an easy task. A comparison based on different models (BDLM, NNs, ARIMA, machine learning algorithm etc.) could lead to the best model for modelling and forecasting COVID-19 in Pakistan. Based on the results of this study, it is proposed to Government of Pakistan to contain the further spread of the virus and reduce the daily new infections/cases through diligent strict measures across the country.

## Supporting information

**S1 Data.**
(CSV)

## Author Contributions

**Conceptualization:** Firdos Khan.

**Data curation:** Firdos Khan.

**Formal analysis:** Firdos Khan.

**Funding acquisition:** Ramesh Kumar.

**Methodology:** Firdos Khan.

**Project administration:** Firdos Khan.

**Software:** Firdos Khan, Alia Saeed.

**Visualization:** Firdos Khan.

**Writing – original draft:** Firdos Khan.

**Writing – review & editing:** Shaukat Ali, Alia Saeed, Ramesh Kumar, Abdul Wali Khan.

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
