## [Decision Letter · Decision Letter 0]

12 Apr 2021

PONE-D-21-07441

FORECASTING FUTURE SCENARIOS OF COVID-19 BY USING BAYESIAN DYNAMIC LINEAR MODELS IN PAKISTAN

PLOS ONE

Dear Dr. Khan,

Thank you for submitting your manuscript to PLOS ONE. After careful consideration, we feel that it has merit but does not fully meet PLOS ONE’s publication criteria as it currently stands. Therefore, we invite you to submit a revised version of the manuscript that addresses the points raised during the review process.

The manuscript requires further revisions regarding at least motivation, contributions, results, and discussion.

We look forward to receiving your revised manuscript.

Kind regards,

Stefan Cristian Gherghina, PhD. Habil.

Academic Editor

PLOS ONE

Journal Requirements:

"No fund received. "

4. Please include your tables as part of your main manuscript and remove the individual files. Please note that supplementary tables should be uploaded as separate "supporting information" files.

Reviewers' comments:

Reviewer's Responses to Questions

**Comments to the Author**

1. Is the manuscript technically sound, and do the data support the conclusions?

Reviewer #1: Yes

Reviewer #2: Yes

Reviewer #3: Partly

2. Has the statistical analysis been performed appropriately and rigorously? 

Reviewer #1: Yes

Reviewer #2: Yes

Reviewer #3: No

3. Have the authors made all data underlying the findings in their manuscript fully available?

Reviewer #1: Yes

Reviewer #2: Yes

Reviewer #3: No

4. Is the manuscript presented in an intelligible fashion and written in standard English?

Reviewer #1: Yes

Reviewer #2: Yes

Reviewer #3: Yes

5. Review Comments to the Author

Reviewer #1: The authors of the paper describe their proposed approach for Forecasting COVID-19 in Pakistan with Bayesian Models. The topic is interesting and with possible applicability. However, the paper needs several improvements:

1) the main contribution and originality should be explained in more detail

2) the motivation of the approach with Bayesian Models needs further clarification, why not other models, like NNs?

3) discussion of related work in COVID-19 should be expanded with more recent work

4) Minor grammar and syntax issues need correction

5) more simulation results and formal comparison of results are needed

6) the conclusions should be extended with more future work

7) More references to COVID-19 papers should be included, like:

Multiple ensemble neural network models with fuzzy response aggregation for predicting COVID-19 time series: the case of Mexico. Healthcare 2020;8:181.

Modeling COVID-19 epidemic in Heilongjiang province, China, Chaos Solitons Fractals, 138, 1–5.

Modeling and forecasting of epidemic spreading: the case of Covid-19 and beyond. Chaos Solitons Fractals 2020;135:109794.

Forecasting of COVID-19 time series for countries in the world based on a hybrid approach combining the fractal dimension and fuzzy logic. Chaos, Solitons and Fractals 140 (2020) 110242

A Novel Method for a COVID-19 Classification of Countries Based on an Intelligent Fuzzy Fractal Approach. Healthcare 2021, 9, 196

Reviewer #2: Comments:

1. By the time authors receive this review, they will have obtained real data for the predicted values of new cases, deaths, and recovered cases. It is suggested to validate the predicted values with the real values.

2. The paper heavily focuses on the quantitative analysis only and does not analyze how and to what extent measures, including the closure of international borders, lockdowns, etc. helped to control infection trends in different countries? A short discussion could be insightful.

3 . For some instances the predicted value is higher and for other instances, it is lower than the real values. One of the reasons for the ambiguity could be a change in control policy and action. The authors are suggested to explain the ambiguity.

Reviewer #3: The merit of the article lies in the production of forecasts for Pakistan's situation in a given period with Bayesian Dynamic Linear Models, with a longer horizon than previous articles.

However, there are several points that need attention. I will first list the main specific ones, and at the end, the general ones.

Specific comments:

- Previous related articles in the introduction could be mentioned in a more detailed critical way, that is stating their merits and drawbacks in a clear way.

1. The value of the forecasts cannot be judged without having other models as benchmarks. At the very least there should be some naive benchmarks.

2. The forecast precision is judged with graphical means without any numerical backing, for which plenty of error metrics exist. Plots do not reveal the detail about the forecast errors.

3. For the benefit of the reader it would be good to be explicit about how forecasts are produced for different horizons (1, 2, ... , 20 days).

4. Lines 58-61: Add references to back the statements. Be explicit about what "mechanistic models" are. Do you mean state-space models? Do you mean compartmental epidemiological models?

5. Lines 115 and 116: Name the variables in such a way that cases, deaths and recoveries are distinguished clearly from each other. Does COV_t represent all of them? It is also helpful to illustrate the dependencies of states. See for example https://www.mdpi.com/2036-7449/13/1/27/htm or

https://onlinelibrary.wiley.com/doi/epdf/10.1002/sim.2566

6. Equations 1 and 2: The details of F_t and G_t are not shown. What is their formulation for your model?

7. Lines 140-141: The dependencies in your model should be made explicit. In them, the variables should be clearly included. COV seems to be a package or an aggregation of variables (infections, deaths, recoveries). Is formulation of dependencies in lines 140 and 141 generic, or the actual dependencies in your model?

8. Section 3.3: It is not clear how the dependency structure of the model (which was not clearly added) was constructed or validated. An article that shows construction via cross-correlation and validation via examining posterior probabilities is the following:

https://onlinelibrary.wiley.com/doi/epdf/10.1002/sim.2566

9. Lines 165-167: The reasons for the choice should be stated.

10. Lines 301-302: Maximum and minimum figures do not correspond with the wording.

You surely mean: minimum 2,132; maximum 4,149

11. About the data: WHO is cited (lines 54 and 55), but the citation directs to

https://www.worldometers.info/coronavirus/

There must be more clarity on which data is taken from which source.

General comments:

Overall, the writing of the article seems rushed. This is understandable in the current situation for many researchers. But for the benefit of the readers and the scientific community, it is important to clearly assert the decisions made in the research, supporting them with graphic means if feasible and adding enough detail of the models. To this purpose, the authors can look at previous articles of the same type. Careful writing can also improve the quality of the work. Expressions such as "[...] forecast is a tricky-subject [...]" should be articulated in a more helpful way.

The main criticism to the study is that, while it aims at forecasting scenarios, those are not included. Scenarios are different situations that can be modelled and assessed through different means. For example, by varying the underlying assumptions, or the initial conditions of the model; or by assuming different initial populations in some models; or by considering the presence or absence of government interventions. The way scenarios are formulated varies, partly depending on the type of model, and there is ample freedom to do this. In the article they are not explicitly included.

A sample study with scenarios determined by how authors modelled epidemic growth can be found at https://www.mdpi.com/2077-0383/9/2/523

This one defines scenarios based on the basic reproductive number, R0:

https://www.medrxiv.org/content/10.1101/2020.03.16.20036939v1.full.pdf

This one explores scenarios based on changing the proportion

of infected with respect to other groups of people:

https://journals.plos.org/plosone/article?id=10.1371/journal.pone.0230405

The absence of scenarios, having aimed the article at it, is a major issue. On this ground, the article, in my view, should be rejected. One possibility for the authors could be to submit their work under a different title, not including scenarios. Even in that case, the other comments provided are applicable, which would lead to major changes, aimed at demonstrating that the research is of a high standard. The sample publications mentioned in the comments might be helpful.

6. PLOS authors have the option to publish the peer review history of their article (what does this mean?). If published, this will include your full peer review and any attached files.

Reviewer #1: **Yes: **Oscar Castillo

Reviewer #2: No

Reviewer #3: No

---

## [Author Response · Author response to Decision Letter 0]

7 May 2021

The authors are extremely grateful to the editor and reviewers to spare time form their busy schedule to review our manuscript.

---

## [Decision Letter · Decision Letter 1]

4 Jun 2021

FORECASTING DAILY NEW INFECTIONS, DEATHS AND RECOVERY CASES DUE TO COVID-19 IN PAKISTAN BY USING BAYESIAN DYNAMIC LINEAR MODELS

PONE-D-21-07441R1

Dear Dr. Khan,

We’re pleased to inform you that your manuscript has been judged scientifically suitable for publication and will be formally accepted for publication once it meets all outstanding technical requirements.

Kind regards,

Stefan Cristian Gherghina, PhD. Habil.

Academic Editor

PLOS ONE

Additional Editor Comments (optional):

Reviewers' comments:

Reviewer's Responses to Questions

**Comments to the Author**

1. If the authors have adequately addressed your comments raised in a previous round of review and you feel that this manuscript is now acceptable for publication, you may indicate that here to bypass the “Comments to the Author” section, enter your conflict of interest statement in the “Confidential to Editor” section, and submit your "Accept" recommendation.

Reviewer #1: All comments have been addressed

Reviewer #3: All comments have been addressed

2. Is the manuscript technically sound, and do the data support the conclusions?

Reviewer #1: Yes

Reviewer #3: Yes

3. Has the statistical analysis been performed appropriately and rigorously? 

Reviewer #1: Yes

Reviewer #3: Yes

4. Have the authors made all data underlying the findings in their manuscript fully available?

Reviewer #1: Yes

Reviewer #3: Yes

5. Is the manuscript presented in an intelligible fashion and written in standard English?

Reviewer #1: Yes

Reviewer #3: Yes

6. Review Comments to the Author

Reviewer #1: The authors have made all the suggested changes and have addressed all my concerns. In my opinion, the paper deserves publication.

Reviewer #3: (No Response)

7. PLOS authors have the option to publish the peer review history of their article (what does this mean?). If published, this will include your full peer review and any attached files.

Reviewer #1: No

Reviewer #3: No

---

## [Editor Report · Acceptance letter]

9 Jun 2021

PONE-D-21-07441R1 

Forecasting daily new infections, deaths and recovery cases due to Covid-19 in Pakistan by using Bayesian Dynamic Linear Models 

Dear Dr. Khan:

I'm pleased to inform you that your manuscript has been deemed suitable for publication in PLOS ONE. Congratulations! Your manuscript is now with our production department. 

Kind regards, 

on behalf of

Dr. Stefan Cristian Gherghina 

Academic Editor

PLOS ONE